# Pd/Alumina Catalysts for Beneficial Transformation of Harmful Freon R-22

Monika Radlik [1], Wojciech Juszczyk [2,*], Erhard Kemnitz [3,*] and Zbigniew Karpiński [1,*]

1   Faculty of Mathematics and Natural Sciences, College of Exact Sciences, Cardinal Stefan Wyszyński University in Warsaw, ul. Wóycickiego 1/3, PL-01938 Warszawa, Poland; m.radlik@uksw.edu.pl
2   Institute of Physical Chemistry, Polish Academy of Sciences, ul. Kasprzaka 44/52, PL-01224 Warszawa, Poland
3   Institut für Chemie, Humboldt-Universität zu Berlin, Brook-Taylor-Str. 2, 12489 Berlin-Adlershof, Germany
*   Correspondence: wjuszczyk@ichf.edu.pl (W.J.); erhard.kemnitz@chemie.hu-berlin.de (E.K.); z.karpinski@uksw.edu.pl (Z.K.)

**Abstract:** Chlorodifluoromethane (R-22), the most abundant freon in the atmosphere, was subjected to successful hydrodechlorination in the presence of palladium supported on γ-alumina, at a relatively low reaction temperature (180 °C). The combination of catalytic actions of alumina (performing freon dismutation) and Pd nanoparticles (catalyzing C–Cl bond splitting in the presence of hydrogen) results in the transformation of freon into valuable, chlorine-free products: methane and fluoroform, the mixture of which is used in plasma etching of silicon and silicon nitride. Very highly metal dispersed $Pt/Al_2O_3$ catalysts, with metal particles of ~1.3 nm in size, are not as effective as $Pd/Al_2O_3$, resulting in only partial dechlorination. A long-term dechlorination screening (3–4 days) showed good catalytic stability of Pd/alumina catalysts.

**Keywords:** $Pd/Al_2O_3$; effective hydrodechlorination of freon R-22; time-on-stream catalytic behavior

## 1. Introduction

Hydrochlorofluorocarbons (HCFCs) have been selected as the first, albeit still provisional, substitution products for the strong ozone-depleting chlorofluorocarbons (CFCs). Since 2015, it has been illegal in the European Union to use any HCFCs to run refrigeration and air-conditioning equipment [1]. The Montreal Protocol and its 2007 amendment targeted a 97.5–100% reduction of the total production of HCFCs by 2030 for all countries. Today, R-22 is by far the most abundant HCFC in the atmosphere, so it is imperative to continue to monitor the evolution of its atmospheric concentration [2–5]. East and South-East Asia have been a substantial source of R-22 emission for a number of years [5]. It is also imperative to note the significant use of $CHClF_2$ in the manufacture of polytetrafluoroethylene (Teflon$^{TN}$), the process potentially involved in R-22 emission. Huge stocks of R-22 must be destroyed or, preferably, converted to other valuable chemicals. The hydrodechlorination (HDC), a catalytic exchange of chlorine by hydrogen appears to be one of the most prospective technologies for the beneficial utilization of harmful chlorine-containing wastes by transforming them into useful, nontoxic products [6–13].

However, because of the low reactivity of $CHClF_2$, much higher reaction temperatures must be used than in the case of hydrodechlorination of other chlorine-containing compounds, like $CCl_2F_2$ [14]. It is difficult to activate the C–Cl bond in the $CHClF_2$ molecule. Its bond dissociation energy is 355 kJ/mol, much more than in the $CCl_2F_2$ molecule with the C–Cl bond energy of 320 kJ/mol [10]. $CHClF_2$ hydrodechlorination over supported palladium catalysts has been the subject of several studies. A short survey of published results on HDC of $CHClF_2$ by differently supported palladium catalysts is grouped in Table S1 (Supplementary Materials). Some numerical data were gathered from graphs, thus they may be somewhat inaccurate. Nevertheless, these data allow to withdraw several

conclusions from this compilation. First, as mentioned before, the low reactivity of $CHClF_2$ has forced researchers to investigate its hydrodechlorination at either higher temperatures or at higher contact times. Second, it is seen that the reaction carried out at higher temperatures generally results in lower selectivity to the targeted $CH_2F_2$ because consecutive reaction toward unwanted methane becomes dominating. Palladium deposited on activated carbons and aluminum fluorides catalysts were most frequently investigated in this reaction, whereas the catalytic behavior of alumina-supported palladium (and nickel) was only reported in [13]. It was shown that rather high reaction temperatures (from 300° to 450 °C) are needed to obtain high $CHClF_2$ conversions. The last point concluded from Table S1 is that supported Pd catalysts experience a considerable deactivation during $CHClF_2$ hydrodechlorination [10]. The difficulty in the activation of the C–Cl bond can be overcome by transforming this unreactive molecule into Cl/F exchange reaction products: $CHCl_3$ and $CHF_3$. This reaction, known as dismutation, can be easily performed on Lewis acid catalysts, such as halogen-pretreated aluminas [6,7]. As a result of the very fast dismutation reactions $CHF_3$ and $CHCl_3$ are finally formed. The latter is significantly more reactive in hydrodechlorination reactions, thus, it can be hydrodechlorinated on metallic centers present in the system resulting in an almost complete conversion into methane [7]. Verification of this idea is the subject of the present study. The main focus was to obtain full dechlorination of R-22 at the lowest possible reaction temperature. To this aim, the C–Cl bond hydrogenolysis activity of two metals: palladium and platinum in combination with the dismutation performance of γ-alumina was studied. It appeared that it was possible to obtain nearly full dechlorination of R-22 at 180 °C, with methane and fluoroform as dominant products, i.e., the mixture used in plasma etching of important materials for microelectronics [15–18].

## 2. Results and Discussion

Data on the physical characterization of reduced 1 wt% $Pd/Al_2O_3$, 1 wt% $Pt/Al_2O_3$ and two $Pd\text{-}Pt/Al_2O_3$ originate from our previous work [19] and are repeated in Table 1. After reduction in hydrogen at 400 °C for 17 h, the catalysts showed a rather high level of exposed metal fractions, with metal particle sizes of ca. 2 nm, as measured by TEM (Transmission Electron Spectroscopy). STEM-EDX (Scanning Transmission Electron Microscopy with Energy-Dispersive X-ray Spectroscopy) data showed a relatively good degree of Pd–Pt alloying in $Pd\text{-}Pt/Al_2O_3$ catalysts prepared from metal acetylacetonates. Reasonable information about the surface composition of Pd–Pt alloy phases was also assessed from catalytic screening in n-hexane conversion [19] and were in line with STEM-EDX results. This relation, implying a high surface enrichment in palladium, matched the expected relationship between the surface and bulk compositions of Pd–Pt alloys [20], see also SET S1 (Supplementary Materials).

**Table 1.** Metal dispersions and metal particle sizes of alumina-supported Pd, Pt and Pd-Pt catalysts.

| Catalyst [a] | Metal Dispersion [b] (D), % | Metal Particle Size, nm | | |
|---|---|---|---|---|
| | | $d_{TEM}$ [c] | $d_{XRD}$ [d] | $d_{chem}$ [e] |
| $Pd/Al_2O_3$ | 28 | 2.5 | 3.3 | 4.0 |
| $Pd80Pt20/Al_2O_3$ | 23.4 | 2.6 | n.m. [f] | 4.8 |
| $Pd50Pt50/Al_2O_3$ | 33.4 | 2.4 | n.m. [f] | 3.4 |
| $Pt/Al_2O_3$ | 44.5 | 1.3 | <2 [g] | 2.5 |

[a] Designation as in Experimental. [b] Based on hydrogen chemisorption, taken from [19]. [c] Metal particle size measured by Transmission Electron Microscopy, taken from [19]. [d] Calculated with the Scherrer equation using (111) reflection, after subtracting background from $Al_2O_3$ (Figure S1). [e] Based on the relation $d_{Pd}$ (nm) = 112/D, from Ichikawa et al. [21], and $d_{Pt}$ (nm) = 113/D, from Rachmady and Vannice [22]. [f] Not measured. [g] Invisible XRD lines from Pt (Figure S1).

Table 1 also contains the results of newly conducted XRD (X-Ray Diffraction) studies which have confirmed the presence of very strongly dispersed platinum and less dispersed

palladium phases. These conclusions stem from Figure S1 (Supplementary Materials) which shows a featureless difference metal profile from 1 wt% $Pt/Al_2O_3$ (part B) and well determined (111) and (200) reflections from palladium (part A).

$Pd/Al_2O_3$, $Pt/Al_2O_3$ and two bimetallic $Pd-Pt/Al_2O_3$ samples were tested in the reaction of $CHClF_2$ with hydrogen in a relatively short preliminary kinetic runs after catalysts reduction at 400 °C for 17 h. The results of these investigations are presented in Figures 1 and 2. Bearing in mind that the desired high rate of $CHClF_2$ dismutation on the alumina support is obtained after its halogenation at higher temperatures [6,7], our experiments began with a high temperature, i.e., 300 °C. After obtaining a relatively constant conversion of $CHClF_2$ (within $\pm 10\%$) at 300 °C, the reaction temperature was gradually decreased to 140 °C.

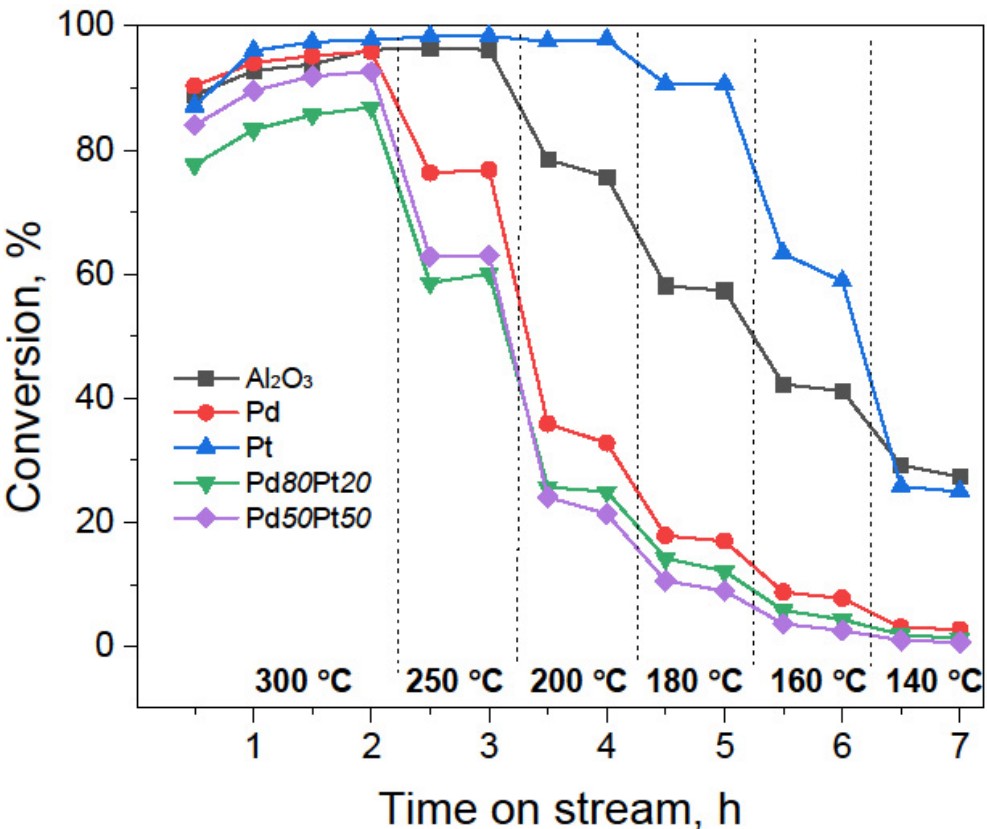

**Figure 1.** Effect of reaction temperature on the overall conversion of freon R-22 over alumina, $Pd/Al_2O_3$, $Pt/Al_2O_3$ and two $Pd-Pt/Al_2O_3$ catalysts. Operating conditions: $CHClF_2/H_2/Ar$ = 1:8:39, GHSV = 14,400 $Ncc/hg_{cat}$, $P \approx 1$ bar.

Since at the same time of the reaction the obtained dismutation products are expected to be hydrogenolyzed to methane on metal (Pd, Pt) centers, the purpose of preliminary kinetic runs was to obtain information on the lowest reaction temperature at which such chlorine removal is effective. These results showed that the best results are obtained for Pd-containing catalysts, where, at 180 °C chlorinated methanes are not observed in the reaction products (see Table 2 and further discussion on selectivity variations). This level of reaction temperature has been used in a series of further long-term studies to be presented later, after discussion of the results of preliminary runs.

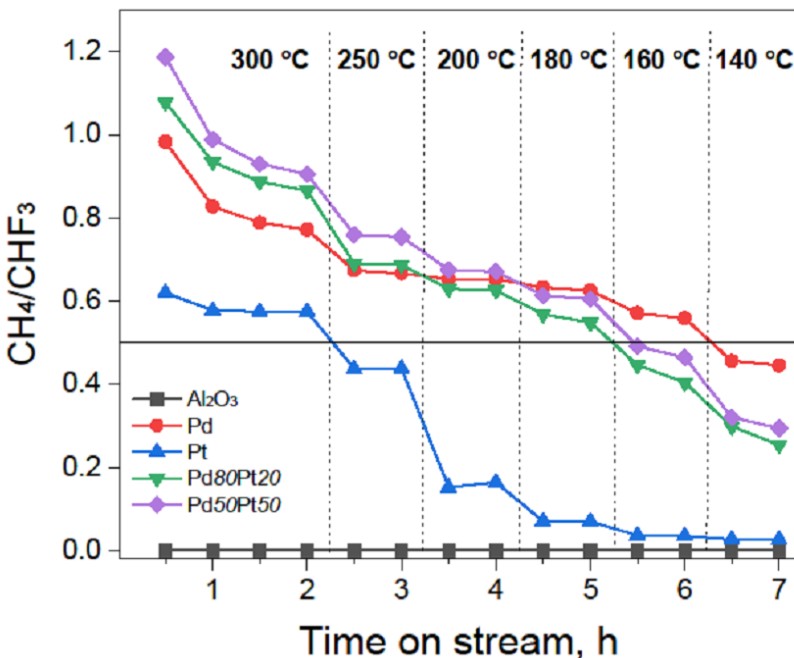

**Figure 2.** Effect of reaction temperature on the $CH_4/CHF_3$ product ratio in the hydrodechlorination of freon R-22 ($CHClF_2$) over alumina, $Pd/Al_2O_3$, $Pt/Al_2O_3$ and two $Pd-Pt/Al_2O_3$ catalysts. Note that the $CH_4/CHF_3$ product ratio $\geq 0.5$ (resulting from total hydrogenolysis of $CHCl_3$, marked by black solid line) would be an indication of full dechlorination of R-22. Operating conditions as in Figure 1.

**Table 2.** Hydrodechlorination of freon R-22 ($CHClF_2$) over alumina-supported metal catalysts: extents of conversion and product selectivities at the reaction temperature 180 °C. Data obtained in preliminary kinetic runs. Operating conditions: $CHClF_2/H_2/Ar = 1:8:39$, GHSV = 14,400 Ncc/hg$_{cat}$, P $\approx$ 1 bar.

| Product Selectivity, % | $Al_2O_3$ | $Pd/Al_2O_3$ | $Pd80Pt20/Al_2O_3$ | $Pd50Pt50/Al2O3$ | $Pt/Al_2O_3$ |
|---|---|---|---|---|---|
| $CH_4$ | - | 37.7 | 36.4 | 32.0 | 3.7 |
| $CHF_3$ | 60.4 | 59.6 | 59.9 | 62.2 | 61.0 |
| $C_2H_6$ | - | 0.9 | 0.5 | 0.7 | 0.3 |
| $CHFCl_2$ | 6.3 | 1.8 | 3.2 | 5.0 | 3.0 |
| $CCl_3F$ | 0.1 | - | - | - | - |
| $CH_2Cl_2$ | - | - | - | - | 2.6 |
| $CHCl_3$ | 33.1 | - | - | - | 29.3 |

The dual-function catalytic character of alumina-containing metal catalysts results from the presence of two types of active sites: Lewis acid active centers in halogen-containing alumina and surface metal sites. The latter sites are capable of dissociating hydrogen and splitting the C–Cl bond, leading to substitution of Cl by H in the organic molecule.

Cl/F exchange or dismutation reaction is known to occur on Lewis acid sites. For example, a high surface $AlF_3$ as very strong solid Lewis acid is highly effective in this reaction [6,7]. $\gamma$-$Al_2O_3$ shows low initial reactivity at 250 °C, then conversion of $CHClF_2$ increases to 96% after 2 h on stream. After cooling the reactor, the catalytic activity started to decline and the conversion level of $CHClF_2$ remained 39% at room temperature [6]. This means that the high dismutation activity of $\gamma$-$Al_2O_3$ is positively shaped by the halogen-containing products of the reaction. Dismutation of $CHClF_2$ leads to two main products: fluoroform ($CHF_3$) and chloroform ($CHCl_3$), see below. It also proceeds on Lewis acid sites, coordinatively unsaturated Al sites, present in highly preheated aluminas. In addition, initial contact with halogen-containing freons makes the surface of alumina even more

acidic due to $AlF_3$ formation offering stronger Lewis acid sites. Patil et al. [7] proposed the following reaction sequence for $CHClF_2$ disproportionation:

$$2\ CHClF_2 \rightarrow CHFCl_2 + CHF_3 \tag{1}$$

$$2\ CHFCl_2 \rightarrow CHCl_3 + CHClF_2 \tag{2}$$

$$5\ CHClF_2 \rightarrow 3\ CHF_3 + CHFCl_2 + CHCl_3 \tag{3}$$

Equation (3) indicates that the $CHF_3/CHCl_3$ product ratio should be 3. However, in our work, only small quantities of $CHFCl_2$ were observed in the reaction products (Table 2). This indicates that the second reaction (2) is fast and goes nearly to completion. In such a case, the presence of $CHFCl_2$ in the first reaction (1) would be replaced by the products of the second reaction ($CHCl_3$ and $CHClF_2$). This simplifies the overall reaction to:

$$3\ CHClF_2 \rightarrow CHCl_3 + 2\ CHF_3 \tag{4}$$

Thus, the $CHF_3/CHCl_3$ product ratio should be 2 if dismutation is the only reaction and no $CHFCl_2$ is present in reaction products. This is what one expects for the reaction occurring on purely acidic catalysts (like halogenated aluminas). However, the appearance of novel metal centers shifts this ratio to higher values, because chloroform ($CHCl_3$) is easily removed by the reaction with hydrogen [23,24]:

$$CHCl_3 \xrightarrow{H_2} CH_2Cl_2,\ CH_3Cl,\ CH_4,\ HCl \tag{5}$$

The complete transformation of chloroform to methane leads to the $CH_4/CHF_3$ ratio = 0.5. Primary experiments with alumina, $Pd/Al_2O_3$, $Pt/Al_2O_3$ and two $Pd\text{-}Pt/Al_2O_3$ catalysts showed their catalytic performance at the temperature range 140–300 °C. Figure 1 shows the evolution of overall activity expressed as the overall extent of freon conversion. It is seen that at 300 °C (initial reaction temperature) all five catalysts tend to increase their activity in time. Such activation is ascribed to an increasing Lewis acidity of alumina, associated with its progressive halogenation. The most catalytically active are alumina and $Pt/Al_2O_3$ catalysts, and the gradual decrease in the reaction temperature affects the resulting conversion rate differently. The $Pt/Al_2O_3$ catalyst is always the most active. $Pd/Al_2O_3$ is less active, but bimetallic catalysts have proven to be even less active (Figure 1).

Variations in product selectivity data are presented in Table 2 (for the reaction temperature 180 °C) and Figure 2. For graphic presentation we selected observed changes in the $CH_4/CHF_3$ ratio because, as mentioned earlier, a complete transformation of chloroform (from dismutation) to methane leads to the $CH_4/CHF_3$ ratio = 0.5. A deviation from this value can be expected for two reasons. First, $CHF_3$ species would react with alumina leading to alumina fluorination and an increase in the $CH_4/CHF_3$ quotient. It will be shown that fluorination of alumina at lower reaction temperatures (at 180 °C) requires longer reaction times (*vide infra*). Similarly, a direct hydrodechlorination of freon on metal sites [13]:

$$CHClF_2 \xrightarrow{H_2} CH_4 + (HCl + HF) \tag{6}$$

also increases the $CH_4/CHF_3$ quotient. Although both effects are expected to be more relevant at higher reaction temperatures, e.g., at 300 °C, the use of this quotient is nevertheless limited in this work to a qualitative comparison of the behavior of three catalysts: $Al_2O_3$, $Pd/Al_2O_3$ and $Pt/Al_2O_3$. Accordingly, the zero value of this quotient for novel metal free $Al_2O_3$, clearly shows that this catalyst is only active in dismutation of R-22. The presence of novel metal in $Pt/Al_2O_3$ slightly increases this parameter, up to ~0.6 at 300 °C, just slightly above the expected theoretical value of 0.5. However, its values decrease at lower reaction temperatures (to 0.16 at 200 °C and 0.07 at 180 °C), suggesting a diminishing contribution of metallic platinum in the HDC of $CHCl_3$. Therefore, this catalyst is less effective in R-22 dechlorination. Three palladium-containing catalysts are better in this

respect. The $CH_4/CHF_3$ ratio about 0.5 is still observable at the reaction temperatures 160–180 °C. This indicates that palladium-containing catalysts, especially $Pd/Al_2O_3$, are superior in total dechlorination of freon at relatively low reaction temperatures.

Theoretical analysis of the hydrodechlorination of chloromethanes on palladium catalysts showed that the rate of reaction correlates well with the C–Cl bond energy, suggesting that the scission of this bond is rate-determining [23,24]. Thus, the efficiency of this step should be also related to the metal-chlorine bond strength. Because Pt is a more noble metal than Pd, both the reactants as well as reaction intermediates (like chlorine) should be less strongly bonded to the surface of platinum. Accordingly, Deeth and Jenkins [25] determined Pd-Cl and Pt-Cl bond energies in $[PdCl_6]^{-2}$ and $[PtCl_6]^{-2}$ as 1739 and 1551 kJ/mol, respectively. Similarly, Erley found that the activation energies of chlorine desorption from Pt surfaces are considerably lower than the respective values characteristic for Pd single crystals: 199 versus 253 kJ/mol for (111) planes [26] and 249 versus 272 kJ/mol for (110) planes [26,27]. Therefore, one can conclude that the fact that platinum nanoparticles are not as effective as palladium, contributing very little to complete dechlorination, should be attributed to the weaker metal-chlorine bond.

However, for comparing the catalytic properties of metals in the reactions of hydrodechlorination of chloromethanes, one should also consider a possible effect of metal particle size. Very small platinum particles embedded in $\gamma$-$Al_2O_3$ (1.5 nm) are very poorly active in $CCl_4$ hydrodechlorination [28]. This effect was attributed to an extensive surface chlorination of very small electro-deficient Pt particles interacting with $\gamma$-alumina [29]. Because our previous TEM studies showed the presence of such small metal particles in the $Pt/Al_2O_3$ (Table 1), one should also expect their poor catalytic performance in the hydrodechlorination of $CHCl_3$, the product of $CHClF_2$ dismutation. Larger metal particles in the $Pd/Al_2O_3$ catalyst ($\geq$2.5 nm, Table 1) should be less charged than 1.5 nm Pt particles in $Pt/Al_2O_3$. This may explain an apparent disagreement with the results of Alvarez-Montero et al. [30,31], who found a better performance of platinum compared to palladium on carbon supported catalysts in $CHCl_3$ hydrodechlorination. It appears that the role of support (alumina or carbon) is decisive here. The cited authors clearly state that the good performance of Pt/C is due to the high proportion of metal in the zero-valent state ($Pt^0$) which disfavors the stabilization of chlorocarbon compounds at the active centers of the catalyst.

Very high dismutation activity of the $Pt/Al_2O_3$, higher than that of bare alumina (Figure 1), may be linked to the presence of electrodeficient Pt species, exhibiting Lewis acidity. However, another factor should be considered here. A considerable color change of alumina, from white to dark beige with black spots, was observed after reaction, suggesting that the alumina is prone to deactivation by deposition of coke deposits. Introduction of Pt to alumina should be beneficial in diminishing catalyst deactivation, by action of hydrogen spillover from metal to the support.

For further kinetic experiments, which were aimed toward testing the catalytic stability of catalysts, we selected only three samples: $Al_2O_3$, $Pt/Al_2O_3$ and $Pd/Al_2O_3$. Long-term experiments at the reaction temperature of 180 °C, lasting 80–90 h, showed quite stable activity of $\gamma$-$Al_2O_3$ (Figure 3a) and an increase in the conversion for $Pd/Al_2O_3$ (Figure 3b). Variations in product selectivity over time have been shown to be minor. Alumina shows only dismutation activity, producing $CHCl_3$ as a major product.

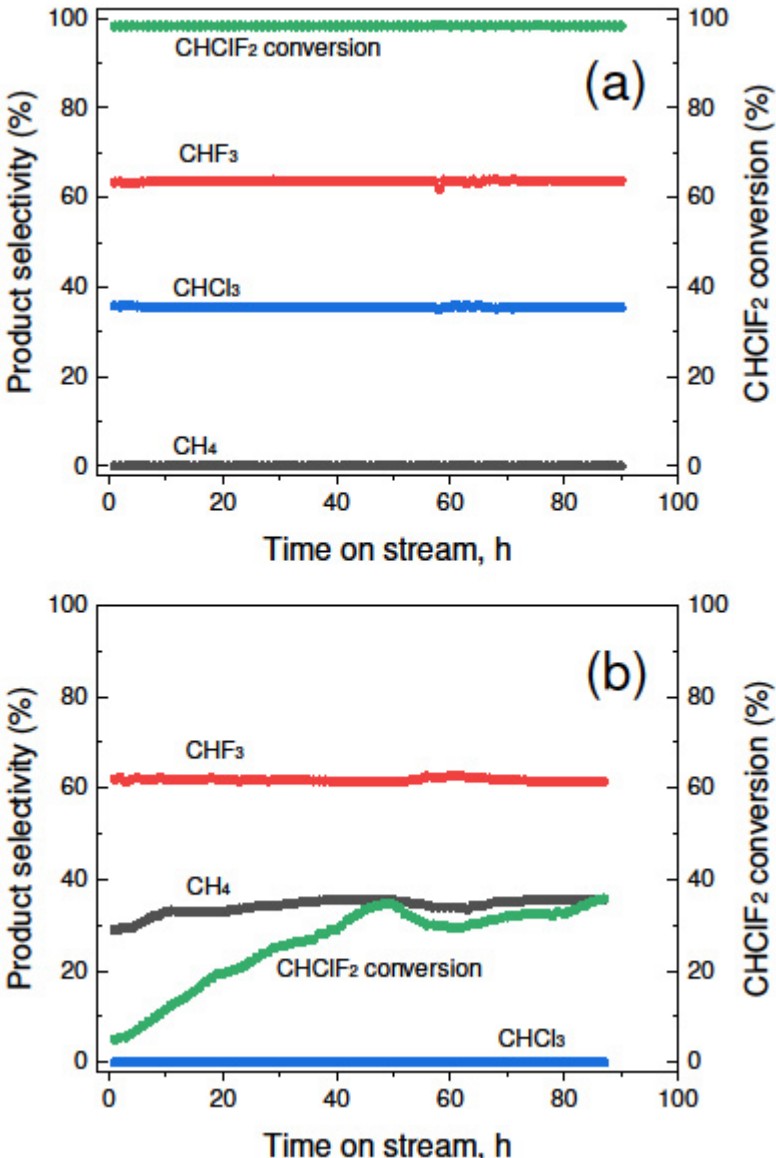

**Figure 3.** Overall conversions and product selectivities in the hydrodechlorination of freon R-22 over: alumina (**a**) and Pd/Al$_2$O$_3$ (**b**) catalysts. Time-on-stream behavior in long-term experiments. Reaction temperature 180 °C. Operating conditions as in Figure 1.

On the other hand, Figure 3b shows an activation of the Pd/Al$_2$O$_3$ catalyst which reaches stable behavior after ~50 h at 180 °C. The increase in the dismutation activity must be linked to a gradual halogenation of alumina. It means that the values of CH$_4$/CHF$_3$ quotient higher than 0.5 shown in Figure 2 result from support fluorination rather than from direct CHClF$_2$ hydrodechlorination (to CH$_4$) on Pd sites. This conclusion is supported by the finding that essentially only CH$_4$ and CHF$_3$ were found in products indicating that dismutation proceeds significantly faster than HDC since otherwise also CH$_2$F$_2$ and CH$_3$F should be present.

However, the most important finding is that nearly total dechlorination (>98%) is observed on Pd/Al$_2$O$_3$, resulting in methane and fluoroform (CHF$_3$) as dominant products (Table 2). Furthermore, this superiority is maintained in long-term experiments (Figure 3). Both chemicals constitute a mixture used in plasma etching of SiO$_2$/Si [15–17] and other materials [18]. Figure 4 shows the stability of two Pt/Al$_2$O$_3$ samples differing in their activation temperature (400 or 500 °C). It appears that the increase in activation temperature which should result in a decrease in metal dispersion (from 44.5% to 41.2%,

Table 2 in [20]) causes only marginal changes both in the total activity and product selectivity. Large amounts of $CHCl_3$ in the product mixture evidently indicates the limited hydrodechlorination activity of $Pt/Al_2O_3$ catalysts. Only a small part of $CHCl_3$ is subject to hydrodechlorination to $CH_2Cl_2$ and methane at 180 °C (Table 2).

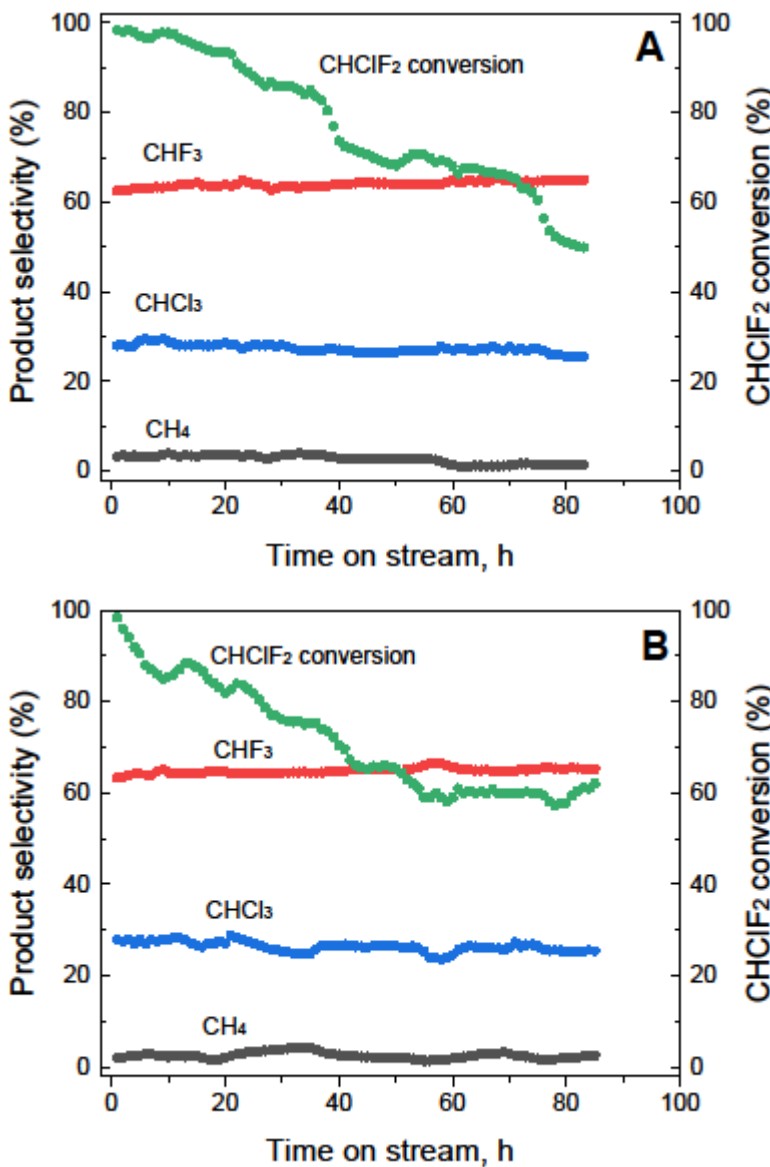

**Figure 4.** Overall conversion and product selectivity in the hydrodechlorination of freon R-22 over $Pt/Al_2O_3$ catalysts: (**A**)—sample prereduced in $H_2$ at 400 °C, (**B**)—sample prereduced in $H_2$ at 500 °C. Time-on-stream behavior. Reaction temperature 180 °C. Operating conditions as in Figure 1.

## 3. Methods

### 3.1. Catalysts

Metal-alumina catalysts preparation and characterization using a variety of methods (Temperature-Programmed Reduction, $H_2$ chemisorption, X-ray Diffraction and Transmission Electron Microscopy) were described in our previous report [19]. Briefly, the Pd/alumina, Pt/alumina and Pd-Pt/alumina catalysts with metal content of 1 wt% were prepared from $\gamma$-alumina (Sasol Puralox Scca, granulation 150–200 mesh, surface area ~ 200 $m^2/g$, from Sasol Germany GmbH, Hamburg, Germany) impregnated with Pd and/or Pt bis-acetylacetonate precursors ($Pd(acac)_2$ and $Pt(acac)_2$, both from Sigma-Aldrich, Saint Louis, MO, USA, purity 99%) dissolved in toluene (analytical reagent from Chempur,

Piekary Śląskie, Poland). In this report, apart from two monometallic $Pd/Al_2O_3$ and $Pt/Al_2O_3$ catalysts, we investigated only two (of five in [19]) bimetallic samples, characterized by atomic Pd/Pt ratios 4:1 and 1:1. These bimetallic catalysts will be designed as in our previous work [19], e.g., Pd*80*Pt*20*, reflecting atomic percentages of Pd and Pt in the metal phase.

### 3.2. Catalyst Characterization

Characterization of Pd-Pt/$Al_2O_3$ catalysts subjected to different reduction conditions was described elsewhere [19]. The catalysts showed a rather high level of exposed metal fractions, with metal particle sizes ca. 2 nm, as measured by TEM. STEM-EDX data showed a relatively good degree of Pd–Pt alloying. Because the characterization was caried out nearly 5 years ago, we decided to verify it for two of the most relevant catalysts: 1 wt% Pd/$Al_2O_3$ and 1 wt% Pt/$Al_2O_3$ catalysts. XRD experiments with reduced Pd/$Al_2O_3$, Pt/$Al_2O_3$ and $Al_2O_3$ catalysts were performed on a standard Rigaku Denki diffractometer (Rigaku Denki, Ltd., Tokyo, Japan) using Ni filtered $CuK_\alpha$ radiation. The samples were scanned in the 2θ range of 30–75° using a step-by-step technique at 2θ intervals of 0.05° and a recording time of 10 s for each step. To improve the sensitivity, the scans were repeated a few times. The resulting diffraction profiles originating from the metal (Pd or Pt) phase were obtained through subtraction of the alumina background profile. The metal crystallite size was assessed from the Scherrer equation using the metal (111) reflection.

### 3.3. Catalytic Screening

The hydrodechlorination of freon R-22 was conducted in a home-made glass flow system equipped with a U-tube reactor with a sintered disc, operating under atmospheric pressure. Prior to each reaction run, the catalyst (0.20 g sample) was subjected to activation in the flow of 10% $H_2$/Ar mixture (20 $cm^3$/min) at 400 °C for 17 h. Next, the samples were cooled in an $H_2$/Ar flow to the desired initial reaction temperature, usually to 300 °C, and exposed to the flow of the reaction mixture containing freon R-22. This step served as pre-halogenation of the $Al_2O_3$ matrix, i.e., to incorporate certain amounts of Cl and F there. For a typical reaction run, the total flow of the reactant mixture was 48 $cm^3$/min and consisted of freon R-22 (1 $cm^3$/min), hydrogen (8 $cm^3$/min), and argon (39 $cm^3$/min). All flows were controlled by Bronkhorst HI-TEC (AK Ruurlo, The Netherlands) mass flow controllers. Typical preliminary reaction runs lasted ~7 h. Additional experiments were conducted to check the stability of the catalytic behavior over the course of 3–4 days. The post-reaction gas was analyzed by gas chromatography (Bruker SCION456-GC, with flame ionization detector, from Bruker-Poland, Poznań, Poland) using a 5% fluorcol/carbosieve column (20 ft) from Sigma-Aldrich (Saint Louis, MO, USA). The results of GC analysis were used to calculate the overall extent of freon R-22 conversion and product selectivities. Product selectivities were defined as $(C_i/\Sigma C_i) \times 100\%$, where $C_i$ is the molar concentration of the detected product i.

## 4. Conclusions

The dual-functional behavior of Pd/alumina catalysts appeared beneficial in catalytic transformation of chlorodifluoromethane (freon R-22), resulting in nearly total chlorine removal (>98%) from the $CHClF_2$ molecule at a relatively low reaction temperature (180 °C). Freon dismutation easily proceeds on halogen-pretreated alumina yielding fluoroform ($CHF_3$) and chloroform ($CHCl_3$) as the dominant products. On the other hand, Pd nanoparticles effectively catalyze $CHCl_3$ hydrodechlorination to methane (Figure 5).

This dual-functional behavior results in the transformation of freon into valuable, chlorine-free products: methane and fluoroform. Both products constitute the mixture used in plasma etching of silicon and silicon nitride. Very small platinum particles (~1.3 nm) appear to contribute greatly to freon dismutation, but are much less effective than palladium in the hydrogenolysis of the C–Cl, contributing very little to complete hydrodechlorination. The course of a long-term kinetic run at 180 °C with Pd/$Al_2O_3$ indicates its initial activation

and the reaching of a relatively stable activity towards dismutation after 50 h of time on stream. This activation must result from the gradual fluorination of the support surface at the initial period of the reaction.

**Figure 5.** Dual-function mechanism of hydrodechlorination of $CHClF_2$ on $Pd/Al_2O_3$ catalysts.

**Supplementary Materials:** The following are available online at https://www.mdpi.com/article/10.3390/catal11101178/s1. Table S1: Compilation of published data on catalytic activity of palladium in hydrodechlorination of chlorodifluoromethane ($CHClF_2$). SET S1: Some data on characterization of Pd-Pt/$Al_2O_3$ catalysts taken from Radlik M., Śrębowata A., Juszczyk W., Matus K., Małolepszy A., Karpiński Z. *n*-Hexane conversion on γ-alumina supported palladium-platinum catalysts, Adsorption 25 (2019) 843–853. Figure S1: XRD profiles of reduced Pd/$Al_2O_3$ (A) and Pt/$Al_2O_3$ (B) catalysts. Difference profiles (green lines) are assumed to originate from metal phases of different crystallinity.

**Author Contributions:** M.R. was responsible for catalysts synthesis and reaction studies, experiment planning and data evaluation; W.J. was responsible for conceptual work, for catalyst characterization using XRD and manuscript writing; E.K. was responsible for conceptual work and manuscript writing; Z.K. was responsible for experiment planning and manuscript writing. All authors have read and agreed to the published version of the manuscript.

**Funding:** This work was carried out within Research Project # 2016/21/B/ST4/03686 from the National Science Centre (NCN), Poland. Partial financial support from the Institute of Physical Chemistry of PAS.

**Data Availability Statement:** The data presented in this study are available in the article and supplementary material here.

**Acknowledgments:** Z.K. acknowledges partial support from the National Science Centre (NCN) Poland within Research Project #2016/21/B/ST4/03686. W.J. is thankful for support from the Institute of Physical Chemistry of PAS.

**Conflicts of Interest:** The authors declare no conflict of interest.

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
