# Peer review of "Pd/Alumina Catalysts for Beneficial Transformation of Harmful Freon R-22"

_catalysts, doi:10.3390/catal11101178_

Round 1

Reviewer 1 Report

The authors have successfully developed a method for hydrodechorination of chlorinated difluoromethane with Pd/Al2O3 catalyst. The systematic study explains the superiority of Pd/Al2O3 over other catalysts screened here for that purpose.

Please consider the following suggestions:

Although the authors have repeated the physical characterisation of catalysts (TEM, STEM-EDX)  from their previous work the data is missing in supporting informations and hence it is needed to check the descriptions. Moreover the instrument details has been used for catalyst characterisation were missing in some experiments.

Overall, the article can be considered for publication. 

Reviewer 2 Report

This manuscript reports a study of the hydrodechlorination of a hydrofluorochlorocarbon. The work comes at a time where there is significant interest in this transformation. There have been numerous previous approaches and the current work is not put in proper context. This clearly needs to be addressed.

The research work seems to have been carried out appropriately, but the results are not clearly laid out. The figures and schemes are of very poor quality and this detracts from the clarity of presentation.

The manuscript needs significant editing prior to being accepted and I would recommend that it be reassessed in more depth once these initial changes have been made.

Round 2

Reviewer 2 Report

The authors have made a few changes to the manuscript but not addressed the main concerns - lack of clarity of presentation when it comes to the results and also poor quality of the figures - to say these are "technical objections" is not an appropriate remark. It is the author's responsibility to present the work in as best a form as possible. As such the manuscript is still not acceptable for publication.
